# Postharvest Application of Acibenzolar-S-Methyl Activates Salicylic Acid Pathway Genes in Kiwifruit Vines

**DOI:** 10.3390/plants12040833

**Published:** 2023-02-13

**Authors:** Tony Reglinski, Joel L. Vanneste, Magan M. Schipper, Deirdre A. Cornish, Janet Yu, Jenny M. Oldham, Christina Fehlmann, Frank Parry, Duncan Hedderley

**Affiliations:** 1The New Zealand Institute for Plant and Food Research Limited, Ruakura, Hamilton 3214, New Zealand; 2The New Zealand Institute for Plant and Food Research Limited, Palmerston North 4410, New Zealand

**Keywords:** induced resistance, elicitor, pathogenesis-related genes, gene regulation

## Abstract

The plant defence inducer Actigard^®^ (acibenzolar-S-methyl [ASM]) is applied before flowering and after fruit harvest to control bacterial canker in kiwifruit caused by *Pseudomonas syringae* pv. *actinidiae*. Pre-flowering application of ASM is known to upregulate defence gene expression; however, the effect of postharvest ASM on defence gene expression in the vine is unknown. In this study, the expression of eight ”defence marker” genes was measured in the leaves of *Actinidia chinensis* var. *chinensis*, ”Zesy002,” and *Actinidia chinensis* var. *deliciosa,* “Hayward,” vines after postharvest treatment with ASM and/or copper. There were two orchards per cultivar with harvest dates approximately three weeks apart for investigating potential changes in responsiveness to ASM during the harvest period. In all trials, postharvest ASM induced the expression of salicylic-acid-pathway defence genes *PR1*, *PR2*, *PR5*, *BAD*, *DMR6*, *NIMIN2*, and *WRKY70*. Gene upregulation was the greatest at 1 day and 7 days after treatment and declined to the control level after 3 weeks. In “Zesy002”, the ASM-induced response was greater at the early harvest site than at the late harvest site. This decline was concomitant with leaf yellowing and a reduction in RNA yield. Effects of postharvest ASM on gene expression did not persist into the following spring, nor were vines conditioned to respond more strongly to pre-flowering ASM application.

## 1. Introduction

The global pandemic of bacterial canker of kiwifruit caused by *Pseudomonas syringae* pv. *actinidiae* (Psa biovar3) seriously affected kiwifruit production worldwide, resulting in severe economic losses [1,2]. Orchard management of bacterial canker relies on the removal of infected canes to reduce inoculum and the application of copper-based products, antibiotics, resistance inducers, and biological control agents, which operate directly or indirectly against the pathogen [3,4]. In kiwifruit, defence against Psa infection is mediated primarily by the activation of the salicylic acid (SA) defence pathway [5]. Acibenzolar-S-methyl (ASM), a functional analogue of SA [6], is the active ingredient in Actigard^®^ (Syngenta, Wilmington, ED, USA), a commercial inducer that is used to control Psa in New Zealand kiwifruit orchards. Up to four foliar applications of Actigard (referred to as ASM from hereon) are permitted in New Zealand kiwifruit orchards per season, with sprays on fruiting kiwifruit vines being restricted to the pre-flowering and postharvest periods to limit the risk of chemical residues [7]. Pre-flowering application of ASM in kiwifruit orchards has been shown to induce transcription of SA-pathway defence genes and to reduce Psa leaf necrosis [8]. However, the effects of postharvest ASM application on defence gene expression in the canopy had not been studied. Understanding the vine response to the postharvest application is important because ASM is recommended at this time to protect fruit stalks and leaf scars from infection by the pathogen [9]. Moreover, the harvest period for kiwifruit spans several weeks, and it is not known whether changes in leaf physiology and metabolic activity over this period affect inducibility.

This study measured the expression of SA-defence-pathway genes in the leaves of *Actinidia chinensis* var. *chinensis,* ”Zesy002” and *A. chinensis* var. *deliciosa,* ”Hayward” after postharvest application with ASM. Gene selection was based on previously published studies [3,8,10] and included transcripts for regulatory proteins (*EDS1A*, *AP2ERF2*, *NIMIN2*, and *WRKY70*), enzymes involved with hormone biosynthesis and homeostasis (*BAD* and *DMR6*), and pathogenesis-related proteins (*PR1*, *PR2*, and *PR5*). The phenomenon described as priming [11] was also studied by measuring the gene expression of young leaves in spring to investigate whether postharvest ASM directly affected gene expression in new leaves and/or conditioned vines for an amplified response to the pre-flowering ASM application.

## 2. Results

### 2.1. Postharvest Application of ASM Induces Defence Gene Expression in Kiwifruit Leaves

#### 2.1.1. Year 1 Orchard Trials 

In 2019, the expression of defence marker genes in “Hayward” was greater in leaves from ASM + Cu-treated vines than in the Cu controls at both sites (Table 1). The most strongly induced genes, *DMR6*, *NIMIN2,* and *WRKY70i*, ranged from 2-fold to 11-fold greater in ASM + Cu-treated vines than in the control. At site A (early harvest) the expression of *PR1* and *PR5* showed a 2-fold increase in ASM + Cu-treated vines compared with the Cu control, whereas at site B (late harvest) *PR1*, *PR2,* and *PR5* increased by 4- to 5-fold at 6 d after ASM + Cu application. In the following spring, gene expression levels were not statistically different between treatments, indicating that effects of postharvest ASM + Cu on gene expression did not persist. Pre-flowering application of ASM + Cu at site A resulted in the upregulation of the same genes as with the postharvest spray. An accidental overspray of trial plots by contract sprayers at site B meant that no data were obtained.

#### 2.1.2. Year 2 Orchard Trials

At the early harvest “Zesy002” orchard (site C), there was a 3- to 6-fold increase in the expression of *DMR6*, *NIMIN2*, and *WRKY70* and a 25-fold increase in the expression of *BAD* at 1 day after ASM + Cu treatment, compared with the Cu control (Table 2). At 7 days post treatment, *DMR6*, *NIMIN2*, and *WRKY70i* were 5- to 6-fold greater in ASM + Cu-treated vines than in the Cu control, whilst *PR1* and *PR5* were 3-fold greater. By 19 days after treatment, there was no significant difference in the gene expression level (*p* > 0.05) between treatments. Following the application of a second postharvest ASM + Cu spray, there was a 10-fold increase in *BAD* and *NIMIN2* expression and a 3- to 7-fold increase in *DMR6* and *WRKY70i* expression after 1 day when compared with the Cu control. After 7 days, *DMR6*, *NIMIN2*, *WRKY70i,* and *PR2* were 2- to 5-fold greater in the ASM + Cu-treated vines than in the Cu control. At the late-harvest “Zesy002” (site D), the expression of *DMR6*, *BAD*, *NIMIN2,* and *WRKY70i* increased by 3- to 11-fold at 1 day after ASM + Cu treatment, compared with the Cu control (Table 2). However, by 7 days post-treatment there was no significant (*p* > 0.05) difference between treatments. No further data were obtained from this site because of a low RNA yield in the leaves collected at 22 days after Spray 1, and at 1 day and 7 days after Spray 2. These leaves were showing signs of yellowing and senescence (Figure 1). In “Hayward”, severe frost damage after fruit harvest at site E (early harvest) resulted in the experiment at this site being discontinued. At site F (late harvest) the expression of *BAD*, *DMR6*, *NIMIN2* and *WRKY70* were upregulated in ASM + Cu-treated vines at 1 day and/or 7 days after treatment, compared with the control (Table 2). By 19 days post-application there was no significant difference between treatments. A second postharvest spray was not applied at site F because of insufficient leaf canopy (<10% fill).

### 2.2. Postharvest ASM Has No Direct Effect on Gene Expression in the following Spring

Gene expression levels in young leaves of “Hayward” in 2019 and 2021, and in “Zesy002” in 2021, did not differ statistically (*p* > 0.05) between those vines treated with ASM + Cu or Cu in the previous autumn (Table 3). This indicates no long-term effect of postharvest ASM on gene expression. Moreover, the postharvest treatment did not affect vine responsiveness to pre-flowering ASM treatment, i.e., no significant (*p* > 0.05) differences in gene upregulation were observed between the ASM–ASM and Cu–ASM treatments. Thus, postharvest ASM treatment did not prime vine inducibility in the following spring. Following pre-flowering sprays, all marker genes except *AP2ERF2* were upregulated in the ASM+Cu-treated vines when compared with the Cu control. Gene upregulation tended to be greater in “Zesy002” than in “Hayward”, and greater in “Zesy002” at site D than at site C (Table 3).

## 3. Discussion

The application of ASM to kiwifruit vines after fruit harvest is recommended to protect fruit stalks and leaf scars from infection by Psa [9], despite poor understanding of defence induction during this period. This study demonstrates for the first time that postharvest ASM induces the upregulation of SA-defence-pathway genes in both “Zesy002” and “Hayward” vines. Vines were sprayed with a tank mix containing ASM + Cu; however, gene upregulation was calculated relative to the copper-treated controls. Hence, it is reasonable to attribute the upregulation to ASM. The most responsive genes, *BAD*, *DMR6*, *NIMIN2*, and *WRKY70i*, are associated with fine-tuning SA homeostasis (*BAD* and *DMR6*) [12,13] and regulating the transcription of pathogenesis-related (PR) proteins (*NIMIN2* and *WRKY70i*) [10]. Specific gene expression patterns varied by orchard and by cultivar, but tended to be greatest at 1 day and 7 days after treatment and were no longer significantly different from control values after 3 weeks. This is consistent with the pattern of gene expression induced by ASM in “Zesy002” and “Hayward” kiwifruit vines during spring [8] and with the transient nature of inducible resistance reported in other plant species [14].

The timing of kiwifruit harvest in commercial orchards is determined by a range of fruit maturity parameters, including soluble solids content, firmness, and flesh colour [15]. The fruit harvest of “Zesy002” is generally several weeks earlier than harvest of “Hayward” vines within the same region [16]. During the postharvest period, an ever-increasing proportion of leaves in the leaf canopy begin to senesce, [17,18] and associated changes in leaf physiology and metabolic activity have the potential to negatively affect defence inducibility [19,20]. In “Hayward” vines in year 1 (2019), the extent of defence gene upregulation by postharvest ASM was similar at both the early- and late-harvest orchards. No comparison between “Hayward” orchards was made in year 2 because of frost damage at one site. However, in “Zesy002”, the gene upregulation by ASM was greater at the early-harvest site than at the late-harvest site, suggesting a decline in vine responsiveness to ASM over time. There was a 3 week gap between fruit harvest at these sites, and an observable difference in canopy quality was observed over this period, with yellowing of the leaves and an associated decline in RNA yield. Leaf yellowing is typically accompanied by a decline in metabolic activity [17,18], and this endorses the kiwifruit industry’s recommendations that postharvest ASM should be applied only “if leaves are still green” [9].

There was no direct effect of postharvest ASM treatment on gene expression in “Hayward” and “Zesy002” vines in the spring; nor did the postharvest sprays prime vines for an amplified response to the pre-flowering ASM spray. In general, the most highly upregulated genes by ASM in spring, as at harvest, were *BAD*, *DMR6*, *NIMIN2,* and *WRKY70i*: the same genes which were the most highly upregulated in the autumn. The upregulation of *PR1*, *PR2,* and *PR5* tended to occur later than that of the regulatory genes, in accordance with their relative position in the SA defence pathway. Differences in the amplitude of gene upregulation between the two “Zesy002” sites and between cultivars may be a function of leaf maturity because this has been shown to affect the response to pre-flowering ASM application [8].

In orchard experiments on Japanese pear (*Pyrus pyrifolia* var. *culta*), postharvest applications of ASM reduced scab lesion development caused by *Venturia nashicola* on the leaves and shoots of ‘Niitaka’ trees by over 80% when compared with controls [21]. Furthermore, in the following spring, pseudothecia formation was largely suppressed in fallen leaves from the ASM-treated trees, thus indicating the potential for the postharvest spray to reduce inoculum potential in spring. In the current study, no measurement of Psa leaf necrosis was recorded after harvest because of the difficulty in distinguishing Psa leaf necrosis from other leaf blemishes during this period. However, gene upregulation correlated with Psa control following pre-flowering application of ASM in orchard vines [8], and so gene expression was considered a suitable proxy for ASM efficacy. It is reasonable to propose that gene upregulation after harvest may precede the synthesis of antimicrobial defences that protect leaf scars from new Psa infections and/or suppress existing Psa populations in the vine. Nevertheless, definitive empirical evidence that postharvest ASM protects vines from Psa infection is still lacking.

## 4. Materials and Methods

### 4.1. Orchard Sites

Experiments were conducted in commercial orchards in the Waikato region of New Zealand on the two most commercially important kiwifruit cultivars in New Zealand: *Actinidia chinensis* var. *chinensis* “Zesy002” (marketed as Zespri™ SunGold™ Kiwifruit) and *A. chinensis* var. *deliciosa* “Hayward” (marketed as Zespri™ Green Kiwifruit). Two orchards per cultivar were selected, with 3–4 weeks separating their respective harvest dates (Table 4).

### 4.2. Treatment and Sampling

After fruit harvest, the vines were treated with Kocide^®^ Opti™ (Cu) or with a tank mix containing Kocide Opti and Actigard (ASM + Cu). Because the trials were conducted in commercial orchards, the inclusion of untreated controls was not permitted. Copper was applied at a concentration of 90 g/100 L after harvest and at 70 g/100 L for the pre-flowering application. ASM was applied at 20 g/100 L regardless of season. Treatments were applied at a spray rate of 1000 L/ha using a pressurised handgun. In 2019, there were five single-vine replicates per treatment. In 2021, ten single-vine plots were sprayed per treatment after fruit harvest to accommodate the priming study, in which five of the vines treated with postharvest ASM + Cu were treated with ASM + Cu in spring and five were treated with copper. Similarly, the postharvest copper control plots were treated with either copper or ASM + Cu in the spring (Table 5). At each timepoint, five leaves of similar phenology were collected within a 1.5 m radius of the main trunk of each replicate vine. Care was taken to select unblemished leaves to minimise potential effects of other biotic or abiotic stresses. Six discs (18 mm diameter) were excised from each leaf using a cork borer. They were then pooled by replicate in a plastic vial and snap frozen in liquid nitrogen. The samples were stored at −70 °C until RNA extraction. At orchard F, “Hayward” leaf samples were collected only up to day 19 after the postharvest spray. At this point approximately 10% of the leaf canopy remained, and so no further sprays were applied.

### 4.3. Gene Expression Analysis

Gene expression was determined using the Plexset^®^ platform from NanoString Technologies Inc. (Seattle, WA, USA), and results were analysed using the nSolver™ 4.0 software (Seattle, WA, USA). Four reference genes and eight target genes were used for the gene expression analysis (Appendix A).

The eukaryotic small ribosomal subunit 40S (*40S*), ubiquitin-conjugating enzyme (*UBC*), glyceraldehyde 3-phosphate dehydrogenase (*GAPDH*), and the protein phosphatase 2 (*PP2A*) genes were used as reference genes. The pathogenesis-related protein family 1 (*PR1*), APETALA2 ethylene responsive factor 2 (*AP2*_*ERF2*), Glucan endo-1,3-β-glucosidase (*PR2*), thaumatin-like protein TG4 (*PR5*), NIM-interacting protein 2 (*NIMIN2*), downy mildew resistance 6 (*DMR6*), WRKY transcription factor 70 (*WRKY70i*), and benzyl alcohol dehydrogenase (*BAD*) were used as target genes. The two 50 bp long probes needed for each gene when using the Nanostring technologies were described previously [8]) (Appendix A) except for *DMR6* and *WRKY70i*, for which the capture probes were:

ACGCCCTCACAATTTTGCTTCAGGACCTCCAAGTCTCAGGCCTACAAGTC and TGGAGGAAATATGGACAAAAGGAGATCCTCAATGCCAAATTTCCAAGGTG and the reporter probes were:

CTCAAGGACGGCAAGTGGATGGCCGTCAAACCCCATCCCAATGCCTTTGT and CTACTTTAGGTGCACACACAAGCCTGATCAAGGTTGCCTAGCAACAAAGC, respectively.

All the probes were synthesised by Integrated DNA Technologies Limited (IDT, Singapore). Total RNA was prepared from approximately 100 mg of ground kiwifruit tissue using the Spectrum Plant Total RNA Kit (Sigma-Aldrich, Auckland, New Zealand), following the supplier’s recommendations. Sample purity and RNA concentrations were determined using a Nanophotometer^®^ (Implen, CA, USA). RNA samples were sent at −80 °C to the Grafton Clinical Genomics of the School of Medical Science, University of Auckland, for processing.

### 4.4. Statistical Analysis

Relative expression data were log^2^ transformed for analysis. For each set of data, the relative expression for each gene was analysed using a linear model with factors for replicate, time, replicate × time, first treatment, second treatment, and the interactions between time and the two treatment factors. Where the effects were significant, the means were compared using least significant differences; comparisons were made within each time. Analysis was performed using Genstat, version 20 (VSNI Ltd., Hemel Hempstead, UK, 2020).

## 5. Conclusions

This study confirms that kiwifruit vines are responsive to ASM after fruit harvest. However, there is no evidence that the effect on gene expression persists into the following spring. The amplitude and duration of the postharvest response to ASM depended on the leaf quality at the time of application. Biologically, this makes sense, because defence activation demands actively metabolising, non-senescing tissue; thus, vines will become less responsive to inducers as canopy health declines. Moreover, there is increasing evidence that defence-related phytohormones (SA, jasmonic acid, and ethylene) play differential roles in regulating leaf senescence [22]. Therefore, more detailed investigation of the cross-regulatory mechanisms between plant defence and leaf senescence are warranted. Further studies are also required to establish if gene upregulation following postharvest application of ASM correlates with a greater resistance to Psa infection during the postharvest period, and whether this affects Psa symptom expression in the following spring. This remains a significant knowledge gap.

## Figures and Tables

**Figure 1 plants-12-00833-f001:**
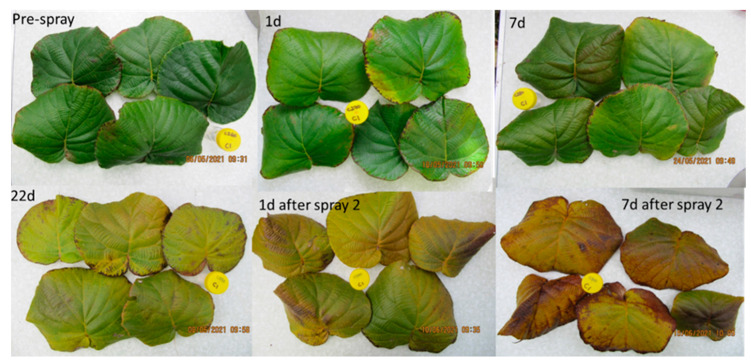
Representative leaves from site D of *Actinidia chinensis* var. *chinensis* “Zesy002” vines in 2021. The leaves were collected from replicate one before the postharvest application (top right) and then at 1 day, 7 days, and 22 days after the first application, and 1 day and 7 days after the second application.

**Table 1 plants-12-00833-t001:** Heat map showing the expression of defence-related genes in *Actinidia chinensis* var. *deliciosa* “Hayward” in 2019. Vines were sprayed with acibenzolar-S-methyl (ASM) after harvest at site A on 8 May and at site B on 24 May. Pre-flowering ASM + Cu at site A was applied on 30 October. Data were log2 transformed for analysis. The numerical values indicate the fold-change relative to the copper control. Fold-changes of two or higher (or 0.5 or lower) were all statistically significant (*p* < 0.05).

2019	Days after ASM	*BAD*	*DMR6*	*AP2ERF2*	*NIMIN2*	*WRKY70i*	*PR1*	*PR2*	*PR5*		

Site A Early harvest	2d	0.9	4.3	0.6	5.0	11.2	1.7	1.0	2.1		Fold change
	6d	1.1	3.9	0.8	1.9	5.2	2.0	0.6	1.6		
Site B Late harvest	1d	3.7	3.9	2.4	5.9	6.9	1.8	1.1	1.1		<2.0
	6d	1.2	6.5	3.0	4.6	5.8	4.7	4.9	4.4		2.0–3.9
Site A Spring	pre-spray	0.9	0.9	0.4	0.6	1.0	1.0	0.9	0.7		4.0–8.0
	2d	1.1	3.9	0.5	4.1	4.4	2.2	1.3	1.8		>8.0
	7d	0.8	2.3	1.7	1.4	1.8	2.3	2.5	3.1		

**Table 2 plants-12-00833-t002:** Heat map showing the expression of selected defence-related genes in *Actinidia chinensis* var. *chinensis* “Zesy002” vines and *A. chinensis* var. *deliciosa* “Hayward” after harvest in 2021. Vines were sprayed with Kocide^®^ Opti™ (Cu) or with a tank mix containing Kocide Opti and Actigard^®^ (acibenzolar-S-methyl (ASM) + Cu). Treatments were applied at site C on 29 April and 20 May, at site D on 17 May and 9 June, and at site F on 3 June. Data were log2 transformed for analysis. The numerical values indicate the fold-change relative to the copper control. Fold-changes of 2 or higher (or 0.5 or lower) were all statistically significant (*p* < 0.05).

2021	Days after ASM		*BAD*	*DMR6*	*AP2ERF2*	*NIMIN2*	*WRKY70i*	*PR1*	*PR2*	*PR5*		
Site C Early harvest ‘Zesy002’	Spray 1	1d	24.8	2.9	1.5	5.3	4.5	1.9	1.0	1.8		
		7d	1.0	6.6	1.0	6.7	5.1	2.8	1.2	3.1		Fold change
		19d	1.0	1.2	0.6	1.0	1.7	0.6	0.7	0.3		
	Spray 2	1d	9.8	3.5	1.5	10.8	7.3	0.6	0.7	0.3		<2.0
		7d	1.2	2.1	1.2	4.8	3.4	1.2	2.4	1.4		2.0–3.9
Site D Late harvest ‘Zesy002’	Spray 1	1d	5.6	3.3	1.5	11.6	5.5	0.9	0.6	0.7		4.0–8.0
		7d	1.3	1.4	0.8	1.3	1.2	1.0	1.9	1.5		>8.0
Site F Late harvest ‘Hayward’	Spray 1	1d	2.3	1.2	1.2	3.1	4.9	1.7	1.2	1.3		
		7d	1.1	2.1	1.2	1.6	2.6	1.3	0.8	1.0		
		19d	1.5	0.8	0.8	1.1	1.1	0.9	0.6	1.1		

**Table 3 plants-12-00833-t003:** Heat map showing the expression of selected defence-related genes in *Actinidia chinensis* var. *chinensis* “Zesy002” and *A. chinensis* var. *deliciosa* “Hayward” kiwifruit vines in spring 2021. Vines were sprayed with Kocide^®^ Opti™ (Cu) or with a tank mix containing Kocide Opti and Actigard^®^ (acibenzolar-S-methyl (ASM) + Cu). Pre-flowering treatments were applied at site C on 11 October, site D on 13 October, and site E on 3 November. Vines sprayed with Cu after harvest were sprayed with either Cu (Cu/Cu) or ASM + Cu (Cu/ASM + Cu) in spring. Similarly, vines sprayed with ASM + Cu after harvest were sprayed with either Cu (ASM + Cu/Cu) or ASM (ASM + Cu/ASM + Cu) in spring. Data were log2 transformed for analysis. The values indicate the fold-change relative to the copper control (Cu/Cu). Fold-changes of two or higher (or 0.5 or lower) were all statistically significant (*p* < 0.05).

Treatment Postharvest/Spring		Days after Treatment	*BAD*	*DMR6*	*AP2ERF2*	*NIMIN2*	*WRKY70i*	*PR1*	*PR2*	*PR5*		
Zesy002’ site C	pre-spray	-	0.9	1.1	1.0	1.3	1.2	1.2	1.1	0.7		Fold change
	Cu/ASM + Cu	1d	5.5	2.3	1.2	1.6	1.3	1.4	0.7	0.8		
	ASM + Cu/ASM + Cu	1d	3.9	2.3	0.8	1.3	1.4	2.0	0.8	1.0		<2.0
	ASM + Cu/Cu	1d	0.5	0.8	0.5	0.6	0.8	0.9	0.8	0.5		2.0–3.9
	Cu/ASM + Cu	7d	1.0	1.5	1.0	1.1	1.0	2.1	1.5	1.8		4.0–8.0
	ASM + Cu/ASM + Cu	7d	0.9	1.6	1.5	1.2	1.0	2.2	1.6	2.0		>8.0
	ASM + Cu/Cu	7d	1.1	1.1	1.8	1.0	1.1	1.4	1.3	1.3		
Zesy002’ site D	pre-spray	-	0.9	1.2	1.1	1.0	1.0	1.2	1.2	1.2		
	Cu/ASM + Cu	1d	8.2	4.0	1.5	8.1	2.4	2.5	1.8	1.7		
	ASM + Cu/ASM + Cu	1d	7.6	4.2	1.9	8.7	2.4	2.5	1.6	1.6		
	ASM + Cu/Cu	1d	1.0	1.6	1.1	1.1	1.3	1.4	1.4	1.5		
	Cu/ASM + Cu	7d	1.3	3.3	1.0	2.0	1.6	5.1	2.4	2.5		
	ASM + Cu/ASM + Cu	7d	1.3	3.3	1.5	2.4	1.7	4.4	2.0	2.3		
	ASM + Cu/Cu	7d	1.1	1.2	1.1	1.3	1.1	1.3	1.2	1.3		
‘Hayward’ site F	pre-spray	-	1.1	1.2	1.1	1.0	1.1	1.1	1.2	1.3		
	Cu/ASM + Cu	1d	3.6	2.9	1.6	3.4	2.7	1.3	1.8	1.6		
	ASM + Cu/ASM + Cu	1d	2.9	3.4	1.3	3.3	2.8	1.4	1.4	1.4		
	ASM + Cu/Cu	1d	1.1	1.0	0.4	0.9	0.9	1.0	1.4	1.4		
	Cu/ASM + Cu	7d	1.0	1.3	0.4	0.8	1.0	1.1	1.0	1.4		
	ASM + Cu/ASM + Cu	7d	1.1	1.7	0.2	0.9	1.3	1.3	1.2	1.7		
	ASM + Cu/Cu	7d	1.1	0.8	0.7	0.8	1.1	1.0	1.0	1.1		

**Table 4 plants-12-00833-t004:** Dates for fruit harvest, applications of Kocide^®^ Opti™ and Kocide^®^ Opti™ + Actigard^®^ and leaf sampling from *Actinidia chinensis* var. *deliciosa* “Hayward” and in *A. chinensis* var. *chinensis* “Zesy002” kiwifruit vines during 2019 and 2021.

			**2019**		**2021**			
	**Cultivar**		**“Hayward”**		**“Zesy002”**		**“Hayward”**	
	Site		A *	B	C	D	E *	F
**Postharvest**	Fruit harvest		6 May	23 May	28 April	15 May	23 June	2 June
	First spray		8 May	24 May	29 April	17 May	24 June **	3 June
	Leaf sampling	1/2 d	10 May	25 May	30 April	18 May	-	4 June
		6/7 d	14 May	30 May	6 May	24 May	-	10 June
		19/22 d	NA	NA	18 May	8 June	-	22 June
	Second spray		NA	NA	20 May	9 June	-	-
	Leaf sampling	1 d	NA	NA	21 May	10 June	-	-
		7 d	NA	NA	27 May	16 June	-	-
**Spring**	Pre-spray leaf sample		29 October	***	11 October	12 October	-	1 November
	Spray application		30 October	-	11 Oct	13 October	-	3 November
	Leaf sampling	1 d	1 November	-	12 Oct	14 October	-	4 November
		7 d	6 November	-	18 Oct	20 October	-	10 November

NA—not applicable (only one spray in 2019). * A and E are different blocks at the same orchard. ** trial discontinued because of severe frost damage. *** trial discontinued because control plots were sprayed with Actigard.

**Table 5 plants-12-00833-t005:** Treatment combinations to *Actinidia chinensis* var. *deliciosa* “Hayward” and in *A. chinensis* var. *chinensis* “Zesy002” kiwifruit vines over postharvest and spring in 2021.

Treatment	Abbreviation	Postharvest	Spring
Control–control	Cu/Cu	Kocide^®^ Opti™	Kocide Opti
Control–Actigard^®^ (ASM)	Cu/ASM ^1^ + Cu	Kocide Opti	Actigard + Kocide Opti
Actigard–control	ASM + Cu/Cu	Actigard + Kocide Opti	Kocide Opti
Actigard–Actigard	ASM + Cu/ASM + Cu	Actigard + Kocide Opti	Actigard + Kocide Opti

^1^ ASM = acibenzolar-S-methyl.

## Data Availability

The data presented in this study are available on request from the corresponding author.

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
