# Peer review of "Postharvest Application of Acibenzolar-S-Methyl Activates Salicylic Acid Pathway Genes in Kiwifruit Vines"

_plants, 2023, doi:10.3390/plants12040833_

Round 1
Reviewer 1 Report
Review of 2207603 Manuscript
Tony Reglinski, Joel Vanneste, Magan Schipper, Deirdre Cornish, Janet Yu, Jenny Oldham, Christina Fehlmann, Frank Parry, Duncan Hedderley:
Postharvest application of acibenzolar-S-methyl activates salicylic acid pathway genes in kiwifruit vines
1. Introduction
The introduction provides an easy-to-follow overview of application of Acibenzolar-S-methyl (ASM), a functional analogue of SA. Are there any among the studied genes that are induced under abiotic stress too? The last sentence of the paragraph is a little ambiguous, whether it is a goal or a result. I suggest you reword it:
The phenomenon described as priming [11] was also studied by measuring the gene expression of young leaves in spring to investigate whether postharvest ASM directly affected gene expression in new leaves and/or conditioned vines for an amplified response to the pre-flowering ASM application .
2. Results
Based on table 5. it is not clear what kind of treatment combination were applied. It is difficult to follow which experimental site x treatment combination the data refer to. I recommend editing the chapter based on description in the material and method.
3. Discussion
in line 162-164: The authors declares: “How ever, in ‘Zesy002’ the gene upregulation by ASM was greater at the early-harvest site than at the late-harvest site, suggesting a decline in vine responsiveness to ASM over time.” In the experiment, the time of the treatments and the sampling were taken the same days after the fruit harvest in the early and late harvest locations. What is the explanation of the expression of different genes in the same variety? Were there any natural infection or abiotic stress, which could be influences the defense gene expression?
In the case of early and late harvest sites, it may be possible to treatment of orchards vines with ASM at different times after harvest resulted maximum effect.
4. Materials and Methods
4.1 Orchard sites: A more precise description of the experimental sites is necessary. It does not mention whether there is a difference in the Psa resistance of the tested varieties. This is important information, because the genetic background of the varieties significantly influences the functioning of the genes responsible for the development of resistance. Were there any natural Psa infection in the experiment?
Table 4 is a bit confused. It is not clear whether ‘Haywand’ sites A and B examined in 2019 are the same as ‘Haywand’ sites E and F examined in 2021. The “Pre-sprey” would be placed in the first column as “Postharvest” was. "Pre-spray" and "Spray application" must be described exactly. I guess that 1/2d, 6/7d and so on are the sampling dates, but it is not obvious. It have to mark it.
4.2 Treatment and sampling: It is not clear from the results whether the treatment combinations presented in Table 5 were applied at all experimental sites.
A few words should also be said about the weather conditions, so that gene induction caused by abiotic stress can be excluded.
The other part of the chapter is sufficiently detailed presentation.
5. Conclusions
It summarizes the results presented in the manuscript well and mentions the possible research directions.
After making the suggested corrections, I recommend publishing the manuscript.

Reviewer 2 Report
In this manuscript, Reglinski et al. describe the transcriptional response of kiwifruit vines to post-harvest applications of the salicylic acid (SA) analogue Acibenzolar-S-methyl (ASM). This treatment was tested in orchard experiments with different kiwifruit varieties at different locations and different time points, and the induction of at least some SA-responsive, defence-associated genes was observed. This induction was transient and did not sensitize plants for enhanced chemically-induced defence gene expression in the subsequent growing season. These data represent an incremental variation on the author's previously published work in Frontiers in Agronomy, which used both potted and orchard kiwifruit plants treated at earlier developmental stages with ASM alone. Nonetheless, the manuscript is adequate for publication following some minor revisions, as described below.
It seems like a more clear picture could have been drawn if the experimental groups included plants with no treatment, ASM alone, Cu alone, and ASM+Cu. Given that copper is known to induce SA signalling genes such as PR1/2/5, the impact of ASM may appear to be small if it cannot further boost the expression of these genes. It is understandable that copper is typically part of the crop protection regimen in kiwifruit, but some additional justification of the experimental design used here is required.
Fig. 1: Are there any data from untreated controls? How much senescence is observed over this timeframe in the absence of chemical treatments (i.e. do the treatments alter the kinetics of senescence)?
Priming is typically considered a conditioned state of readiness to respond to pathogen infection, so the lack of an enhanced response to ASM/Cu in ASM/Cu-treated plants may not be surprising. Reference [9] suggests that post-harvest ASM confers protection in the following growth season, so the classical priming phenotypes may still be applicable. On another note, are the data mentioned in reference [9] available anywhere else, or are they confidential internal data for industry? If available, they could underlie some more definitive statements in the Discussion, especially since the last sentence of the Discussion does not seem to be 100% true.
Typographical corrections:
Line 70: Italicize Latin names
Line 256: Reference formatting
Line 273: log2 transformed
Line 285: omit '?'
Round 2
Reviewer 1 Report
Review of 2207603 Manuscript
Tony Reglinski, Joel Vanneste, Magan Schipper, Deirdre Cornish, Janet Yu, Jenny Oldham, Christina Fehlmann, Frank Parry, Duncan Hedderley:
Postharvest application of acibenzolar-S-methyl activates salicylic acid pathway genes in kiwifruit vines
I suggested the publication of the manuscript.

Reviewer 2 Report
My comments were adequately addressed and the manuscript is suitable for publication.
I am assuming that deleted words/letters were not highlighted by the tracked changes in the manuscript, given that the paragraph starting at line 47 has some extraneous words and sentences. As long as the final version has been proofread, the edits should be fine.